# Oral Health Literacy Levels of Nursing Professionals and Effectiveness of Integrating Oral Health Training into Nursing Curricula: A Systematic Review

**Abdulrhman Albougami** 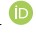

Department of Nursing, College of Applied Medical Sciences, Majmaah University,
Al-Majmaah 11952, Saudi Arabia; a.albougami@mu.edu.sa

**Abstract:** This systematic review assessed the evidence for the oral health literacy levels (i.e., knowledge, attitudes, barriers, oral care and practices, and trainings and resources) of nursing professionals and the effectiveness of integrating oral health training into nursing training. Four electronic databases were searched; however, for relevance, only evidence published between 2013 and 2023 was considered. Overall, 70 studies that focused on five key themes, namely, (i) knowledge of oral healthcare among nurses; (ii) attitudes of nurses towards oral healthcare; (iii) barriers to oral healthcare promotion; (iv) oral care and practices; and (v) trainings and resources to promote oral healthcare, were retrieved. Nurses were found to have a lack of or suboptimal of knowledge regarding oral healthcare. Moreover, their attitudes and practices related to the provision of oral healthcare varied substantially. Key barriers that impeded oral healthcare promotion included a lack of knowledge, awareness, education, skills, and training. Integrating oral health training was considered effective for improving oral health literacy and nurses emphasized the inclusion of such training into their curricula for improving oral healthcare. In summary, nurses have an important role to play in promoting oral health. Furthermore, integration of oral health training into nursing curricula could be a feasible approach to improve the oral health literacy of nurses and reduce the burden of oral disease.

**Keywords:** attitude; barriers; knowledge; oral care and practices; oral health training; nursing professionals



## 1. Introduction

Oral disease affects approximately 3.5 billion individuals across all age groups [1]. The prevalence of oral disease is continuing to increase with increasing urbanization and sedentary lifestyles, resulting in remarkable health and economic burdens [2]. The impact of oral disease extends beyond oral pain and discomfort, often affecting the functioning and quality of life of the affected individuals [3].

Oral health is an integral part of general health and wellbeing [4]. Evidence suggests that oral health status and oral hygiene practices have a significant role in improving overall health [5]. Nursing professionals account for most of the healthcare workforce and are at the forefront of promoting oral hygiene. Studies have emphasized that nursing professionals do have a prominent role in creating awareness about oral health and imparting integrated oral care [6–8]. However, poor oral health literacy levels, i.e., lack of knowledge, negative attitudes towards oral healthcare, and inadequate training of nursing professionals, have been identified as the major barriers to quality oral healthcare [9].

Integrating oral health training into the learning curricula of nursing professionals would empower them to improve care for individuals and reduce oral-health-related burdens. Previous systematic reviews have indicated that the integration of oral health education into nursing practice improves the quality of oral health and reduces inequities associated with oral health [8,10]. In a systematic review of the literature published between 2008 and 2019, Bhagat et al. emphasized that the incorporation of oral health education

into nursing curricula (via interprofessional education) could be useful to increase the awareness of nursing students, and to improve access to and the ability to provide quality oral care to the elderly population [8]. Similarly, another systematic review assessed the effectiveness of integrating maternal and children's oral health promotion into nursing and midwifery practice. The findings from the review reported that the incorporation of oral health promotion had the potential to reduce oral health disparities and expand access to preventive dental care within poor and disadvantaged communities [10]. Therefore, the present systematic review was conducted to understand the updated evidence on the oral health literacy levels (i.e., knowledge, attitudes, barriers, oral care and practices, and trainings and resources), and to assess the effectiveness of integrating oral health training into the learning curricula of nursing professionals.

The specific research questions of this systematic review were:

1. To understand the knowledge and attitudes of nursing professionals towards providing oral healthcare.
2. To understand the perceived barriers towards oral healthcare promotion. Additionally, the general practices related to oral care and practice were explored.
3. To identify the existing trainings and resources, understand the perception towards incorporating oral health training into nursing curricula, and determine the effectiveness of integrating oral health training into the training of nursing professionals.

## 2. Methods

### 2.1. Literature Search

This systematic review was carried out in accordance with the preferred reporting items for systematic reviews and meta-analyses (PRISMA)guidelines [11]. A systematic search of four electronic databases (i.e., EMBASE™, MEDLINE®, MEDLINE®-In-Process, and Cochrane library) was performed from database inception to 16 February 2023 using the Ovid™ platform. The detailed search strategy used to retrieve published studies is provided in Supplementary Table S1. The evidence from the database searches was supplemented by a grey literature search. Additionally, the bibliographies of the included systematic reviews were examined to identify any additional potential studies.

### 2.2. PRISMA Protocol (Study Selection and Data Extraction)

All retrieved citations were initially screened for relevance based on the title and abstract (i.e., primary screening) by two independent reviewers using predefined eligibility criteria (Table 1). Citations that did not meet the predefined eligibility criteria were excluded, and those with insufficient or unclear information were included in the full-text screening (i.e., secondary screening). The full-text articles of the potentially eligible citations were retrieved and subjected to secondary screening by two independent reviewers. Any conflicts between these two reviewers at the primary and secondary screening stages were reconciled by mutual discussion. Multiple publications of a study were linked together and extracted as a single study.

**Table 1.** Study eligibility criteria.

| | Inclusion Criteria | Exclusion Criteria |
|---|---|---|
| Population | • All nursing professionals | • Non-nursing professionals<br>• Other healthcare professionals |
| Intervention | • Preventive healthcare services (educational, screening) by nurses<br>• Programs in clinics, hospitals (government and private), or households | • Preventive healthcare services (educational, screening) solely driven by other non-nursing professionals<br>• Programs in settings other than those under the inclusion criteria |
| Comparator | • Not applicable | • Not applicable |

**Table 1.** *Cont.*

| | Inclusion Criteria | Exclusion Criteria |
|---|---|---|
| Key outcomes | • Knowledge of oral healthcare among nurses<br>• Attitude of nurses towards oral healthcare<br>• Barriers towards oral healthcare promotion<br>• Oral care and practices<br>• Trainings and resources to perform oral healthcare | • Outcomes other than listed under inclusion criteria |
| Study design | • RCTs<br>• Cluster RCTs<br>• Quasi-experimental studies<br>• Observational studies | • Abstracts<br>• Editorials/commentaries/review articles/expert opinions |
| Language | English | Non-English |
| Search timeframe | Inception to present (February 2023) | Not applicable |
| Any other criteria | No limits on sample size or countries | Not applicable |

Abbreviations: RCTs, randomized controlled trials.

Data from the eligible studies were extracted by a single reviewer into a predefined extraction grid, which was validated for completeness and accuracy by the second reviewer. Any disagreements/ambiguities were resolved through mutual discussion. The key parameters extracted included study information (author name, publication year, study design and objective, and country), baseline data (number of participants and sample size), and the outcomes of interest (i.e., knowledge of oral healthcare among nurses, attitude of nurses towards oral healthcare, barriers towards oral healthcare promotion, oral care and practices, and trainings and resources to perform oral healthcare). Meta-analysis could not be conducted as data from the included studies were presented descriptively.

*2.3. Quality Assessment of the Included Studies and Two-Step Review*

The methodological quality of the included studies (i.e., qualitative, quantitative, mixed, non-randomized, and RCTs) was assessed using the Mixed-Methods Appraisal Tool (MMAT) [12]. The MMAT was selected because it enables the methodological appraisal of a plethora of study designs. Studies were screened for the clarity of the research question and whether the collected data addressed the research question. Each domain of assessment was rated as "yes", "no", or "unclear". The studies that passed through these questions were quality assessed. Studies that met all the assessment criteria were given a rating of 1, and those which did not meet the criteria were rated <1 [12]. Two reviewers assessed the quality of the included studies. Any conflicts between the two reviewers were resolved through mutual discussion.

**3. Results**

Overall, 1409 unique citations were retrieved through electronic database searches. After primary screening, 1121 citations were excluded as they did not meet inclusion criteria, and 288 citations were included in the full-text review and assessment. Of these, 67 records were included in the systematic review. Additionally, 3 studies were included from bibliographic searches. Thus, 70 unique studies were included in the final synthesis. Figure 1 illustrates the selection process according to the PRISMA flow diagram.

Supplementary Table S2 summarizes the characteristics of the included studies. Most of the included studies were from Asia (n = 21), followed by North America (n = 15), Europe (n = 11), the rest of the world (n = 10), the Middle East (n = 9), and Africa (n = 7). While the majority of the studies were of nurses (n = 45) and nursing students (n = 23), two studies were of both nurses and nursing students (n = 2). Of the 70 included studies, the majority were cross-sectional studies (n = 54), followed by observational studies (n = 15) and RCTs (n = 1). The sample sizes across the included studies ranged from 12 [13] to 1888 [14].

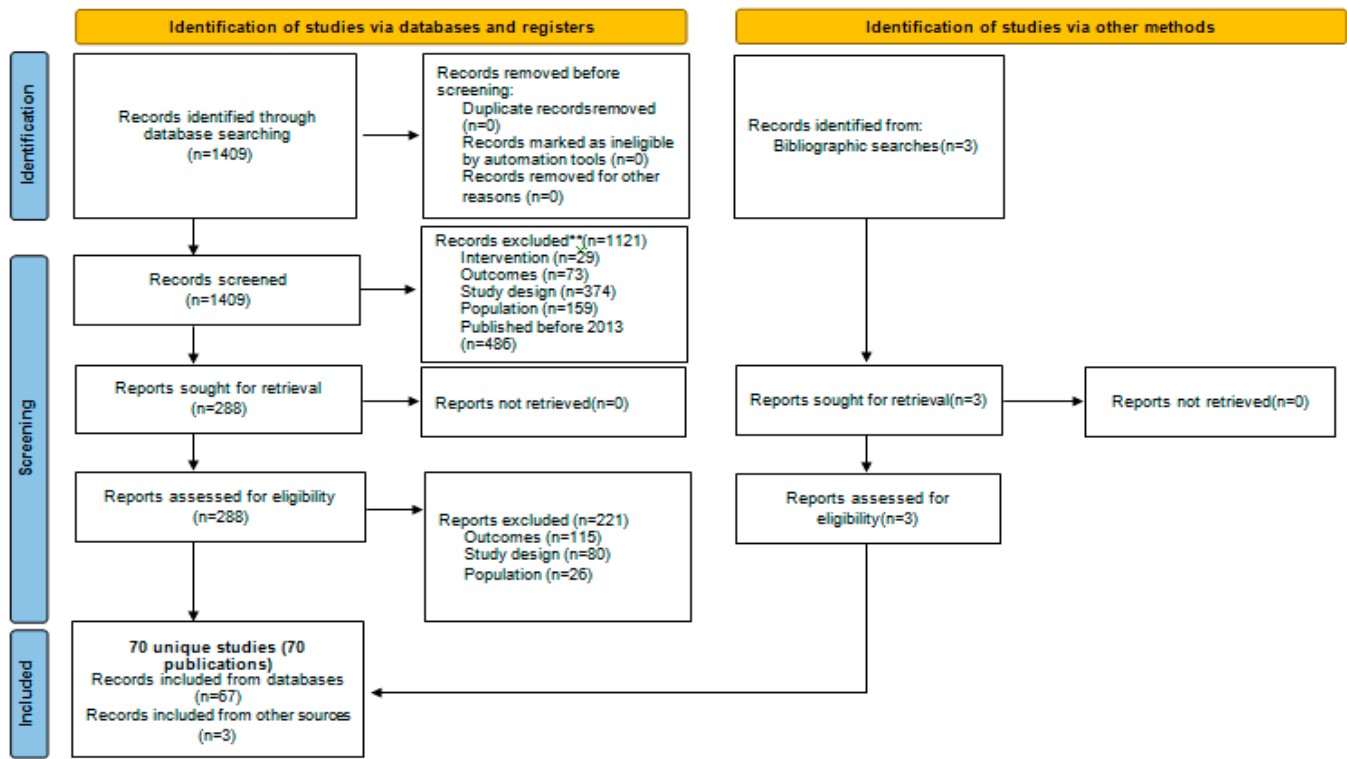

**Figure 1.** PRISMA flow diagram of study selection.

Given the enormous size of the dataset, this study focused on summarizing the evidence published in the last 10 years (i.e., 2013 to 2023). Overall, five important themes were identified from the retrieved evidence, namely, (i) knowledge of oral healthcare among nursing professionals (n = 28); (ii) attitudes of nursing professionals towards oral healthcare (n = 31); (iii) perceived barriers to oral healthcare promotion (n = 12); (iv) oral care and practices (n = 12); and (v) trainings and resources to promote oral healthcare (n = 19) (Supplementary Table S2).

*3.1. Knowledge of Oral Healthcare among Nursing Professionals*

Ten studies reported the effects of implementing oral care programs, interdisciplinary/interprofessional collaborative practice, or educational programs on the knowledge of nurses [14–23]. In general, the implementation of such programs was found to enhance the knowledge of nurses [14–23] (Figure 2). One non-randomized intervention trial evaluated the effect of an oral healthcare program in nursing homes on the knowledge of and attitudes towards oral health among care staff. The oral health program, which included mobile dental care, was found to significantly increase the knowledge of and attitudes towards oral health among care staff in both the intervention and control groups (both $p < 0.001$), with the largest increase observed in the intervention group ($p < 0.001$) [14]. In a questionnaire-based study, oral health literacy initiatives were found to increase the mean oral health literacy score from pre- to post-test (patients: 33.5%, community/parish members: 22.3%, nursing students: 20.8%, and medical students: all $p < 0.0001$) [15]. In a Nigerian study that assessed the impact of an interdisciplinary educational intervention on nurses' knowledge of perinatal and infant oral healthcare, the mean knowledge scores of the nursing practitioners on oral hygiene, teething, trauma, caries, and oral habits were improved after the intervention and 6 months after the educational intervention compared with baseline [19]. Furthermore, interdisciplinary/interprofessional collaborative practice and educational programs were reported to improve the confidence levels and behaviors of nurses related to the promotion of oral health [17,18,23–27].

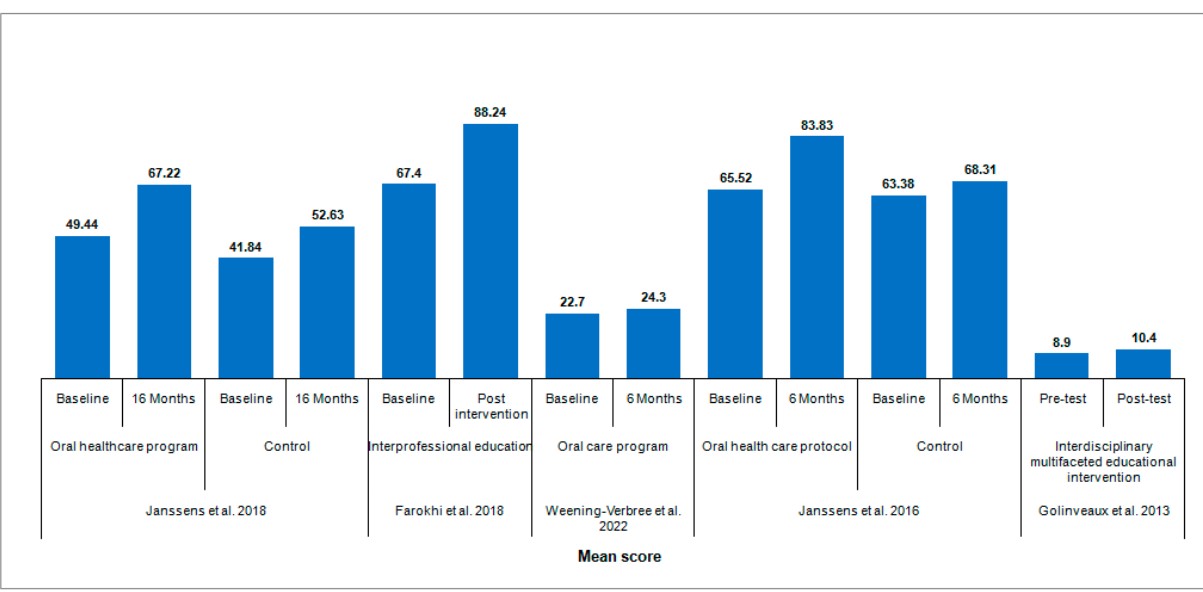

**Figure 2.** Effect of intervention on oral health knowledge [14–16,21,23].

Twenty-seven studies reported on the knowledge of oral healthcare among nursing professionals [13–23,28–43]. Most studies reported (n = 13) suboptimal or a lack of knowledge of oral healthcare among nurses [28–39]. Overall, 53.1% [28] to 100% [35] of nurses reported poor knowledge about oral health (Supplementary Figure S1). The mean oral health knowledge scores among nurses ranged from 3.7 (out of a maximum score of 5.0) [34] to 6.74 (out of a maximum score of 22.0) [39]. Knowledge related to oral healthcare was also reported to be linked to the educational background and experience of the nurses, with nursing practitioners and graduates having relatively better knowledge than trainee/student nurses, midwives, nurses with a diploma, and other professionals [29,33,36–38,40–42]. In a cross-sectional study, nurses with a degree or above were 5 times more likely to have greater knowledge versus diploma holders (adjusted odds ratio (AOR) = 5.05; 95% confidence interval [CI]: 1.85–13.83) [33]. University hospital nurses reported greater knowledge of medication effects than nurses from regional hospitals ($p < 0.001$). Similarly, experienced nurses were reported to have a significantly higher knowledge of medications' effects on oral health than less experienced nurses ($\geq 4$ vs. $< 4$ years of experience; $p = 0.048$) [36]. In comparison to community health officers and community health extension workers, nurses were found to have better knowledge of oral health [38]. Two cross-sectional studies that surveyed the knowledge of undergraduate nursing students reported that most of them indicated a lack of knowledge on oral health topics (periodontal conditions, dental examination, oral hygiene procedure, and oral health intervention methods), as these were not taught or included in their educational curricula [13,29]. A cross-sectional study investigated the current levels of oral-health-related content in undergraduate nursing education in New Zealand. Despite the inclusion of the majority of oral health topics, the most important topics (i.e., risk factors associated with dental caries, periodontal health, and risk factors) were not taught in approximately 40% of the schools [13]. In another cross-sectional study, the knowledge of the relationship between oral–systemic disease and the examination or screening of the oral cavity among nursing graduates was found to be poor, owing to the lack of oral healthcare content in nursing curricula [29]. In general, female nurses were found to possess better knowledge about oral healthcare than males [43] (Figure 3).

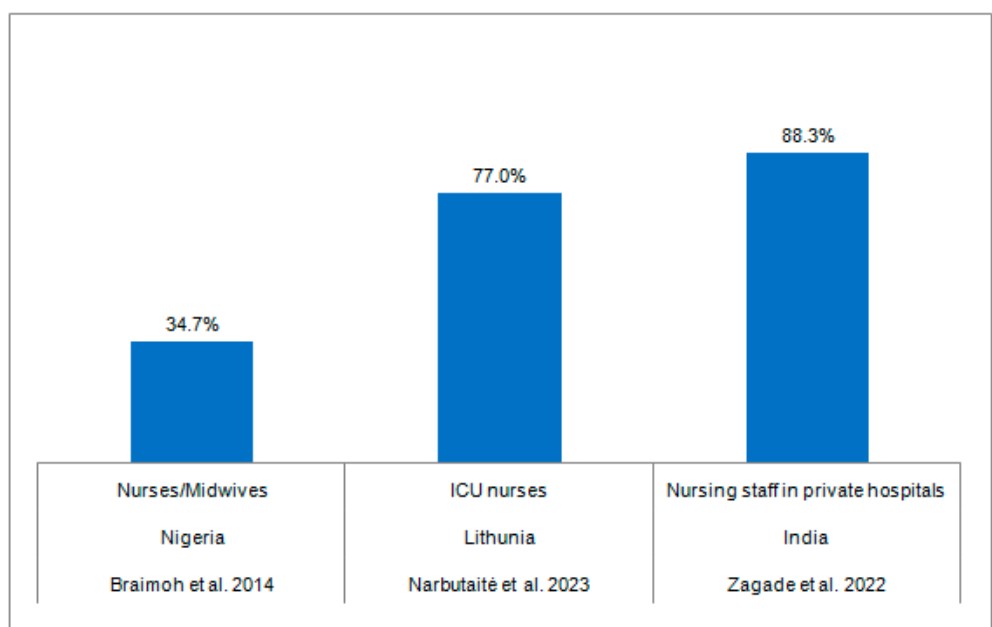

**Figure 3.** Adequate training in oral health [38,44,45].

*3.2. Attitudes of Nursing Professionals towards Oral Healthcare*

The implementation of oral care programs and interdisciplinary/interprofessional/multi-professional programs was reported to improve the attitudes of nurses towards oral healthcare [14,16,18,21,23,24,46] (Figure 4). A non-randomized intervention trial evaluated the effect of an oral healthcare program in nursing homes on knowledge and attitudes regarding oral health among care staff. The results showed that the oral healthcare program, which included mobile dental care, significantly improved the attitudes of the staff towards oral healthcare [14]. Similarly, the implementation of an oral care program was found to improve the knowledge and attitudes of home care nurses towards oral healthcare and its impact on older people's oral health [16]. In a questionnaire-based study from Japan, the attitudes and confidence levels of nursing students were found to significantly improve after an interprofessional education program. Specifically, the total mean scores on the oral assessment ability tests of the nursing students increased after the interprofessional education program (first test: 6.80 (standard deviation, SD: 1.33); second test: 6.96 (SD: 1.38); third test: 7.88 (SD: 1.01) points). Furthermore, their performance of oral self-assessment increased significantly from 15.8% at baseline to 32.7% after the program [24].

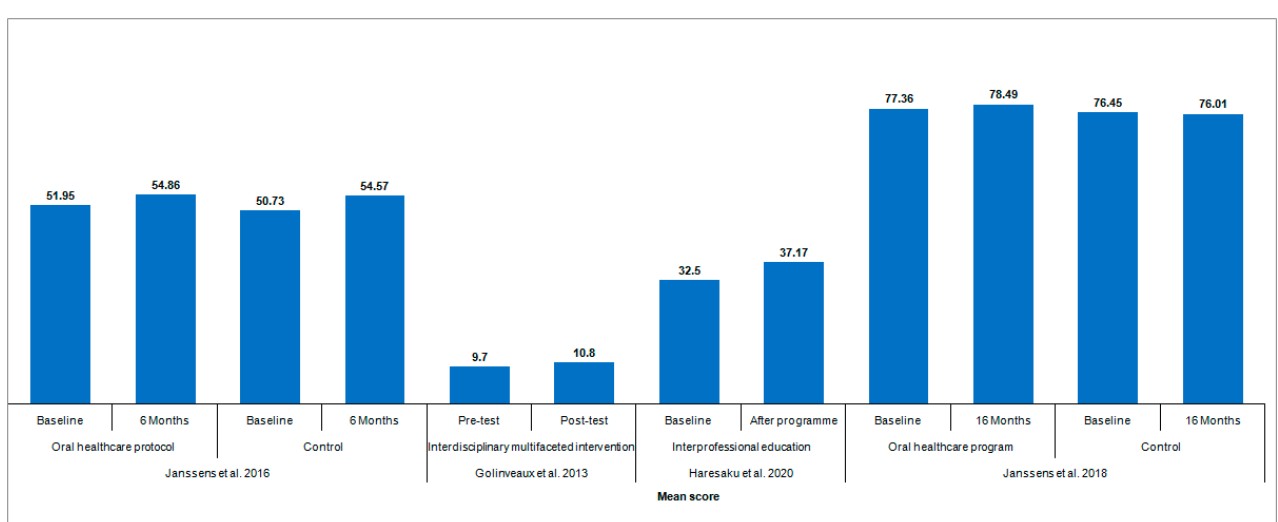

**Figure 4.** Effects of interventions on the attitudes of nurses towards oral health [14,21,23,24].

Thirty studies evaluated the attitudes of nursing professionals towards the provision of oral healthcare [4,14,16,18,21–25,27,33,39,40,43–59]. Evidence from the included studies showed varying attitudes of nurses towards providing oral healthcare. With few exceptions, nursing practitioners and nursing graduates showed more positive attitudes towards the provision of oral healthcare and the prevention and management of oral health problems vs. trainee/student nurse, midwives, nurses with a diploma, and other professionals [27,33,39–41,43,46,49–51,53–55,58,59]. Self-efficacy and attitudes towards providing mouth care were reported to be higher in nurses from public geriatric care facilities than those from private geriatric care facilities, those working in <100 bed vs. >100 bed facilities, and American vs. Chinese nurses [47]. Female nurses reported better attitudes than male nurses (Figure S2 [43]. A Croatian cross-sectional study that assessed the attitudes of 241 ICU nurses showed that they were sufficiently educated about the importance of oral care and the impact of adequate oral care on ICU patients [57]. Similarly, a study by Rabiei et al. reported that nurses were aware of the relationship between dental and general health, and their role in preventing oral disease [58]. Nurses with a degree and above were more likely to have a positive attitude than diploma holders (AOR = 2.01; 95% CI: 1.012–4.01) [33]. Similarly, trained nurses were 3.54 times more likely to have a positive attitude than untrained nurses about oral care (AOR = 3.54; 95% CI: 1.61–7.79) [33]. The mean score of the nurses' attitude towards oral healthcare ranged from 22.0 (out of a maximum score of 32.0) [58] to 40.32 (out of a maximum score 60.0) [39]. Nurses and pharmacists with fewer years of experience (OR = 1.31, 95% CI: 1.43–3.11) and those with higher degrees (OR = 1.4, 95% CI: 1.47–2.27) reported more positive attitudes toward the prevention and management of dental and oral health problems in post-chemotherapy patients than did those with lower qualifications ($p < 0.05$) [49]. In a cross-sectional study that included primary care nurses, higher scores in the pediatric (OR = 1.2) and dental (OR = 1.3) domains and a greater willingness to receive oral healthcare information (OR = 5.3) were associated with a positive attitude toward oral healthcare [58]. Furthermore, nurses with lower educational qualifications (OR = 1.9) and better oral health behavior (OR = 1.1), and those working in non-affluent regions (OR = 1.6), were reported to have more positive attitudes towards oral healthcare versus those with higher educational degrees, those with lower OHB scores, and those working in affluent areas ($p < 0.05$) [58].

### 3.3. Barriers to Oral Healthcare Promotion

Twelve studies reported barriers that impede the promotion of oral healthcare among nursing professionals [4,27,30,31,35,36,57,60–64].The key barriers included lack of awareness (84.4%) [31], knowledge/skills (54.4–76.9%) [30,61], and training (5.9–78.0%) [30,60]; lack of time/low priority (44.0–66.2%) [27,30,36,57,60,63,64]; lack of importance placed on oral care (33.8–74.0%) [27,30,36,61,62];lack of staff/resources (48.5–63.6%) [27,31,57,63,64]; lack of tools/equipment (44.4–91.2%) [30,57,60,61]; and patient behavior [27,31,36,63]. Other barriers reported were work load/schedule (12.8–63.6%) [30,31,35], lack of or limited finances/insurance coverage (70.1–79.2%) [31,63], fear of pain (68.8%) [31], lack of ability to identify dental diseases/mistrust of healthcare professionals (61–70.1%) [31], and logistics (67.5%) [31].

### 3.4. Oral Care and Practices

Six studies reported that the implementation of educational interventions and ventilator-associated pneumonia prevention programs improved oral care [46,65–69]. A Japanese study showed that the percentage of undergraduate nursing students expressing a need to learn techniques for interdental cleaning significantly increased from 21.6–76.5% (at the baseline) to 65.7–95.1% (after the multi-professional education program) both in theory and in practice ($p < 0.05$) [46]. In a Colombian study, an educational intervention aimed at improving oral hygiene care by nursing staff was reported to reduce the incidence of ventilator-associated pneumonia in adults connected to ventilators in the ICU (from 8.9% to 2.8%). Routine oral and dental care by the nursing staff substantially increased from

29.6% to 92.8% after the intervention. In addition to the increase in the oral care among the nurses, the educational intervention also changed the oral practices of the patients (i.e., tooth brushing increased from 0.9% to 48.6%, use of chlorhexidine mouthwash increased from 2.9% to 70%, and aid participation in caring increased from 32.7% to 94.3%) [65]. Similarly, another study from the US reported that the effective implementation of oral care protocols significantly decreased the rates of ventilator-associated pneumonia by 62.5%. The frequency of oral care increased significantly to tooth brushing every 4 h and swabbing every 12 h with 0.12% chlorhexidine solution ($p$ = 0.001) [66]. In a quasi-experimental retrospective study, there was a significant increase in the adherence to preventive measures before and after continuing education/training (i.e., adherence to oral hygiene increased from 89.5% to 98.2% and toothbrushing increased from 80.8% to 96.4% (both $p$ < 0.001)) [67]. In an Iranian study, a formal and structured education program in addition to the usual service significantly improved the performance of nurses in providing oral care for mechanically ventilated children (intervention group: 42.8 (SD: 18.5)) at baseline vs. 68.6 (SD: 31.4) at 4 weeks after the intervention ($p$ < 0.001); control group: 48.7 (SD:15.7) at baseline vs. 48.6 (SD: 15.4)) [68]. Similarly, interpersonal education was found to improve the frequency of tooth brushing, comfort level with dental visits, oral-health-related knowledge, and overall positive responses of nursing and dental students [69].

In addition to assessing the effect of integration of oral care practices into the training of nursing professionals, 12 studies reported on the general oral care practices of nursing professionals [4,44,46,55,57,61,65–70]. In general, there was variability observed in nursing practices related to oral healthcare. Five studies reported that nurses often used oral care solutions (saline, chlorhexidine, hydrogen peroxide solution, sodium bicarbonate solution, and distilled water/tap water) for mouthwash [4,44,57,65,70]. Various studies reported the use of foam swabs/swabbing (61–91.9%) [44,61], moisturizers (53–98.9%) [44,55], cotton/gauze swabs (69.9–93,5%) [55,61], saline (20–82.7%) [61,70], chlorhexidine (12.2–97.8%) [54,57,61,70], hydrogen peroxide solution (31.8%) [61], sodium bicarbonate solution (20.8–75.6%) [61,70], distilled water/tap water (4.3–17.9%) [55,61,70], mouthwash (57.8–88.2%) [55,70], toothbrushes (17.8–70%) [44,70], suction toothbrushes (93.5%) [55], and manual toothbrushes (49.5%) [55] for oral care.

*3.5. Trainings and Resources to Promote Oral Healthcare*

Nineteen studies reported on trainings and resources for promoting oral healthcare [13,14,19,28,30,31,38,44–46,53,55,60,61,63,68,71–73]. Between 34.7% [38] and 88.3% [45] of the nurses perceived that they received adequate training in oral health and the majority felt that the training was optimal [45,46] (Figure 3). Conversely, few studies reported that nurses reported a lack of relevant oral-healthcare-related training [30,60,61,63]. Among nurses who received training, the amount of time devoted to oral health training was limited (<2 h [71] and 5 h [13]). Nurses emphasized the importance of oral-health-related trainings and recommended conducting periodic trainings in future [31,46,55,72]. For example, in a Japanese study, Haresaku et al. reported that nurses who completed a 45 h course over a 4-year period showed increased interest in oral health and intended to collaborate with oral health professionals [46]. In an Iranian study, Behzadi et al. reported that the mean oral healthcare performance scores of nurses improved after receipt of an educational program. Additionally, nurses recommended periodic in-service trainings to enhance their performance in providing oral care [68] (Figure 5). Similarly, another study found that an interdisciplinary education/intervention had a positive impact on the oral health knowledge of nurses as it increased the knowledge of oral care (i.e., hygiene, teething, trauma, caries, and oral habits) among Nigerian nurses [19]. Furthermore, nurses recommended the inclusion of appropriate training and encouragement modules in the curricula of nurse training [14,53].

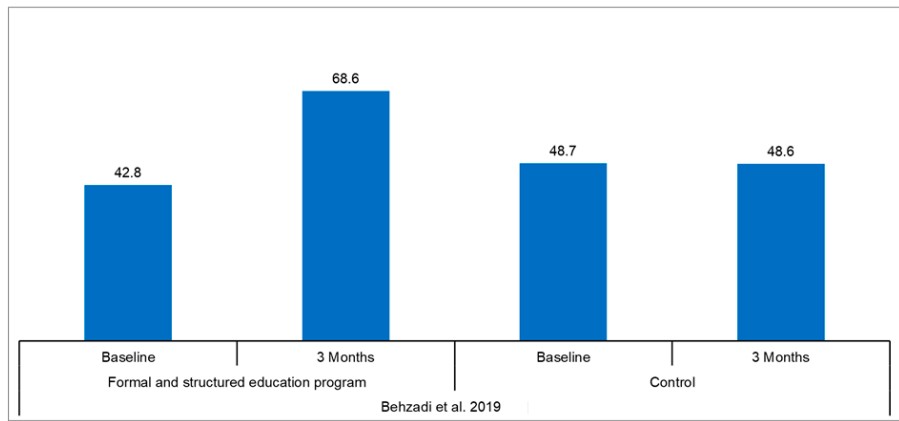

**Figure 5.** Performance scores in oral health [68].

In terms of resources, various methods (scenario simulation analysis, education online, series of lectures, communication and discussion with others, self-study, seminars, workshops, and pamphlets/posters) were reportedly used to train nurses on oral healthcare [28,61]. Nursing textbooks, videos/DVDs, and online materials were the key resources used for training [13,73]. However, the preferred mode of training was seminars (54.3%), followed by workshops (26.0%) [28].

*3.6. Quality Assessment of the Included Studies*

With a couple of exceptions, the overall quality of the majority of the included studies ranged between 0.8 and 1.0 (Supplementary Table S2).

**4. Discussion**

This systematic review provides contemporary evidence on the different components of oral health literacy (i.e., knowledge, attitudes, barriers, oral care and practices, and trainings and resources) among nursing professionals, and underscores the importance of integrating oral health training into nursing curricula to improve oral health outcomes. The results from this review will help nursing professionals to understand and embrace various concepts of oral health literacy, impart quality oral healthcare, and reduce the burden related to oral disease via the integration of oral health training into their learning curricula.

Despite substantial progress, oral conditions/diseases continue to impose a substantial burden on patients, healthcare systems, and society as a whole. Nursing professionals occupy a unique position in the provision of effective oral care and the promotion oral hygiene [8,10,30,33]. Using a systematic approach, this review sought to understand the oral health literacy levels of nursing professionals and assess the effectiveness of integrating oral health education into the learning curricula of nursing professionals.

Overall, five key themes were identified in this review, namely, (i) knowledge of oral healthcare among nursing professionals; (ii) attitudes of nursing professionals towards oral healthcare; (iii) barriers to oral healthcare promotion; (iv) oral care and practices; and (v) trainings and resources to promote oral healthcare. The knowledge and attitudes of the nursing professionals related to oral health were suboptimal/inadequate. Furthermore, there was substantial heterogeneity observed in the attitudes of the nursing professionals towards providing oral healthcare. However, the knowledge and attitudes of nursing professionals were found to be associated with the level of education and experience of the nurses, which is consistent with previous studies that reported higher levels of education leading to higher oral health literacy [33,74,75]. This could be attributed to the fact that increased educational/experience levels could help them to regularly update their knowledge (via participation workshops/seminars/conferences/educational forums) and develop a positive attitude towards providing improved oral healthcare to patients.

Several barriers have been reported to impede the effective promotion and implementation of oral healthcare among nursing professionals [30,31]. Evidence from the current systematic review showed that a lack of knowledge, awareness, education, skills, and relevant trainings were the key barriers often experienced by the nursing professionals in routine nursing care [4,27,30,31,60–63]. Therefore, oral health training or intervention programs targeted to needs and addressing existing barriers could be a feasible approach to improve the current standard of oral care and facilitate sustainable change in oral healthcare. Across the included studies, oral care and practices (in terms of use of oral care solutions) among nursing professionals were found to vary substantially. In line with the published studies, this may be mainly be because of the differences in knowledge, educational/experience levels, attitudes, and perceptions of the nursing professionals related to oral care [76]. Furthermore, the evidence for the adequacy of relevant trainings among the nursing professionals was found to be conflicting. While a few studies perceived that nurses received adequate training in relation to oral health [38,44,46], many studies reported that nurses often received inadequate training [30,60,61,63]. The lack of adequate training could be due to a lack of adequate resources, funding, and time, which needs to be addressed. However, nurses did acknowledge the importance of interdisciplinary education/trainings/interventions in improving oral health literacy and emphasized the inclusion of periodic trainings in curricula for improving oral healthcare. Of the various resources available, seminars (54.3%) and workshops (26.0%) were the preferred modes of training for improving access to and provision of oral healthcare services [28].

Taken together, these findings provide valuable insights on the oral health literacy levels (i.e., knowledge, attitudes, barriers, oral care and practices, and trainings and resources) among nursing professionals. Considering the magnitude of the burden of oral disease, there is a need to enhance literacy levels and alleviate barriers related to oral health among nursing professionals, expand the workforce, and integrate oral health programs into nursing curricula to effectively improve the quality of oral care.

Several caveats of this review merit consideration. First, language bias cannot be excluded because only studies published in the English language were included in this review. Second, this study focused on summarizing contemporary knowledge, attitudes, barriers, oral care and practices, and trainings and resources related to the promotion of oral healthcare (i.e., published in the last 10 years). It is possible that some concepts and practices existing before 2013 could have been missed. Third, the majority of the evidence was derived from cross-sectional studies; cause–effect relationships could not be demonstrated. Fourth, the possibility of potential recall bias cannot be ruled out as the majority of the studies used self-report questionnaires to generate insights. Finally, the high level of heterogeneity across the included studies prevented us from conducting a meta-analysis.

Nevertheless, this review study does have strengths, including the structured and robust methodology employed to conduct this review. A large number of studies were included and analyzed through a systematic method, thus minimizing bias. Furthermore, this systematic review used a standardized quality assessment checklist (MMAT) to appraise the included studies and draw meaningful insights and conclusions on the effectiveness of integrating oral health into the training of nursing professionals.

## 5. Conclusions

In summary, this review suggests that nursing professionals do have an important role to play in promoting oral health. It provides a comprehensive landscape of the oral health literacy (i.e., knowledge, attitudes, barriers, oral care and practices, and trainings and resources) of nursing professionals across the world. Moreover, it emphasizes that the integration of oral health training into nursing curricula could be a feasible approach to improve the oral health literacy of nursing professionals and reduce the burden related to oral disease.

**Supplementary Materials:** The following supporting information can be downloaded at: https://www.mdpi.com/article/10.3390/app131810403/s1, Figure S1. Nurses with poor oral health knowledge [28,30,35]. Figure S2. Knowledge score [43]. Figure S3. Attitude scores [43]. Table S1. Search strategy. Table S2. Characteristics and key findings of the included studies (n = 70) [77–84]. Table S3. Quality assessment of the included studies using Mixed Methods Appraisal Tool [12].

**Funding:** This study received no external funding.

**Institutional Review Board Statement:** Not applicable.

**Informed Consent Statement:** Not applicable.

**Data Availability Statement:** The data will be provided upon request.

**Conflicts of Interest:** The author declares no conflict of interest.

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
