# Peer review of "Oral Health Literacy Levels of Nursing Professionals and Effectiveness of Integrating Oral Health Training into Nursing Curricula: A Systematic Review"

_applsci, doi:10.3390/app131810403_

Round 1

Reviewer 1 Report

Oral health promotion may become one of the important functions of the nurses. The presented systematic review provides the current information regarding oral health literacy of the nursing professionals, attitudes to promote oral health, barriers to oral health promotion, and, finally, trainings and resources to promote oral care.

The author conducted the deep literature search and synthesized the key findings. However, there are several recommendations which may be useful to improve the quality of the manuscript.

1.       Title

The title is Effectiveness of Integrating Oral Health into the Curricula of Nursing Professionals: A Systematic Review. However, the review includes not only this topic, but also other questions related to overall oral health literacy of the nursing professionals, barriers to oral health promotion, trainings and resources to promote oral care, etc.  Therefore, it seems appropriate to rename the manuscript using a broader title.

 2.       Introduction

At the end of the Introduction section please clarify the research question of the systematic review.

 3.       Methods

-          As it was stated in the Introduction section, the study aimed to synthesize updated evidence on the effectiveness of integrating oral health education among nursing professionals. However, in Table 1 the interventions were stated as follows: (1) Preventive healthcare services (educational, screening) by nurses and (2) Programs in clinics, hospitals (government and private) or households. In the supplementary Table 2 (key findings of the included studies) many studies do not assess effectiveness of oral health education in nurses professionals, they just investigate oral health literacy among nurses or attitudes towards oral health care. Please clarify your aim, research question, and inclusion criteria.

 4.       Results

In the results section it would be better to focus on the studies which investigated the efficacy of integration of oral health topics in the educational curricula of nursing professionals. In the current version of the manuscript these studies are just mentioned at the end of each sub-section. It seems to be better to place this information as the primary outcomes at the beginning of each sub-section. Also key numbers should be added. Other results should be either excluded from the manuscript either listed as the secondary outcomes.

Author Response

Dear Reviewer, 

I would like to for all the constructive comments, which have helped me refine the manuscript. Kindly find below the response to the reviewers’ comments. I have made relevant changes to the manuscript. For ease of review, we have annotated the changes in yellow in the revised version of the manuscript.

Thanks and Regards

Abdulrhman Albougami

Associate Professor

Department of Nursing, College of Applied Medical Sciences

Majmaah University, Al-Majmaah 11952

Saudi Arabia

Reviewer 2 Report

1. Minor editing of Introduction section is warranted. Suggestions include:

*Page 1, paragraph 2 - change wellbeing to well-being.

*Page 1, paragraph 2 - change awarenessabout to awareness about

*Create a third paragraph starting with the sentence "Integrating oral health..."

*Also suggest phrasing "Integrating oral health training into formal nursing education..."

2. Methods

*Good thorough description of search strategy. Well designed. However, recommendation to change section 1.2 to PRISMA protocol would be warranted. Authors appear to have followed most of the steps but should clearly state this.  More information can be found at:

Linares-Espinós E, Hernández V, Domínguez-Escrig JL, Fernández-Pello S, Hevia V, Mayor J, Padilla-Fernández B, Ribal MJ. Methodology of a systematic review. Actas Urol Esp (Engl Ed). 2018 Oct;42(8):499-506. English, Spanish. doi: 10.1016/j.acuro.2018.01.010. Epub 2018 May 3. PMID: 29731270.

*Quality assessment can be rephrased as QA and two-step review

*Authors need to modify Methods to include data analysis. Supplementary data files clearly show descriptive statistics and compilation of odds ratio and other systematic review-related data. 

3. Results

*Figure 1 is missing. Not found in the manuscript and not found in the supplementary file. The PRISMA diagram is an essential component of any systematic review. Please revise.

*Suggest the creation of a Forest Plots to more easily describe data from each section. If not, perhaps a histogram to highlight relevant information might help. For example, Section 2.1 states females had better oral healthcare knowledge than males. In which studies? By how much? Graphs would make this easier to digest.

*Sections 2.2 and 2.3 would also be greatly improved by the addition of a graph or forest plot. A good reference guide can be found at:

Andrade C. Understanding the Basics of Meta-Analysis and How to Read a Forest Plot: As Simple as It Gets. J Clin Psychiatry. 2020 Oct 6;81(5):20f13698. doi: 10.4088/JCP.20f13698. PMID: 33027562.

*Section 2.4 needs further clarification. Are these studies looking at procedures that nurses performed that included the oral cavity? If so, then perhaps focusing on the ventilator studies with clear relevance to oral health and nursing outcomes might be more appropriate.

*Section 2.5 is very interesting. Plotting or graphing these data (34% - 77%) by type of study (RN, BSN, LPN) or country (Japan, Nigeria) might also provide more specificity and information for the reader. Plotting the improvements from the in-service trainings or Iranian education program (pre and post) can also be good if the authors can synthesize these data into a forest plot or graph.

4. Discussion and Conclusions

Very thorough and detailed. 

Good. Very minor suggestions.

Author Response

(The authors gave the same response as above.)
